# Analysis of Retinal Blood Vessel Diameters in Pregnant Women Practicing Yoga: A Feasibility Study

**DOI:** 10.3390/healthcare10071356

**Published:** 2022-07-21

**Authors:** Dejan Dinevski, Miha Lučovnik, Ivan Žebeljan, Domen Guzelj, Izidora Vesenjak Dinevski, Adam Saloň, Patrick De Boever, Nandu Goswami

**Affiliations:** 1Faculty of Medicine, University of Maribor, 2000 Maribor, Slovenia; domen.guzelj@student.um.si; 2Department of Perinatology, Division of Obstetrics and Gynecology, University Medical Center Ljubljana, 1000 Ljubljana, Slovenia; miha.lucovnik@mf.uni-lj.si; 3Health center Lenart, 2230 Lenart, Slovenia; ivan.zeblejan@zd-lenart.si; 4Sončna Vila Yoga Studio, 2000 Maribor, Slovenia; izi@izidora.si; 5Physiology Division, Medical University of Graz, 8036 Graz, Austria; adam.salon@medunigraz.at (A.S.); nandu.goswami@medunigraz.at (N.G.); 6Department of Biology, University of Antwerp, 2020 Antwerpen, Belgium; patrick.deboever@uantwerpen.be; 7Centre for Environmental Sciences, Hasselt University, 3590 Diepenbeek, Belgium

**Keywords:** pregnancy, yoga, microcirculation, retinal blood vessels, fundus imaging

## Abstract

Studies have shown that practicing yoga during pregnancy beneficially affects maternal and neonatal outcomes. The objective of this study was to determine the effect of prenatal yoga on the microvascular circulatory system via non-invasive measurements of retinal blood vessels. We included 29 women who practiced prenatal yoga in the study. There were no statistically significant differences in arteriolar and venular diameters pre- vs. post-90 min yoga practice (150.5 ± 11.4 μm pre- vs. 151.2 ± 10.2 μm post-yoga; *p* = 0.53 for arteriolar and 221.8 ± 16.1 μm pre- vs. 223.2 ± 15.7 μm post-yoga; *p* = 0.51 for venular diameters). The current study demonstrated the feasibility of the setup. More extensive studies are needed to determine the potential microvascular effects of practicing yoga throughout pregnancy.

## 1. Introduction

In the absence of obstetric complications or medical contraindications, regular physical activity during pregnancy is recommended due to its numerous proven maternal and fetal health benefits, such as decreased gestational diabetes mellitus, cesarean birth, operative vaginal delivery, postpartum depression, fetal growth restriction, and preterm birth [1]. Over the last few decades, yoga has become a popular form of physical exercise among pregnant women [2]. Yoga is an ancient body–mind practice that covers a system of body postures (asanas), combined with breathing (pranayama) and concentration (dharana) exercises and meditation (dhyana) techniques [3]. Observational studies and randomized trials demonstrated safety and multiple benefits of practicing yoga during pregnancy [4,5,6,7,8,9,10,11]. Yoga can decrease maternal stress, anxiety, sleep disturbances and pregnancy-related lower back/pelvic pain [12,13,14,15,16,17]. It has been associated with shorter duration of labor and lower cesarean delivery rates [5,6,7]. In addition, prenatal yoga can reduce hypertensive disorders in pregnancy (including preeclampsia) and small for gestational age neonatal birth weight [7,8,9,10]. Yoga practices may positively stimulate the function of the circulatory system, which, in turn, may lead to improved placental development and function [18].

Two compartments can be distinguished in the circulatory system: the macrovascular and the microvascular compartment [19]. The macrovascular compartment acts as a buffer system of blood pulsation after left ventricular contraction and ensures a steady blood flow to the microvasculature. The microvascular compartment nourishes all tissues with oxygen and nutrients [19]. The retina is increasingly being used to assess the health status and functioning of the microcirculation. The retinal microvasculature can be visualized directly and non-invasively from an optic fundus picture allowing an accurate and objective assessment of the microvascular compartment [20,21,22]. For example, narrower retinal arterioles and broader retinal venules are associated with an increased risk of cardiovascular morbidity and mortality [23,24,25]. Regular physical activity induces retinal arteriolar dilation and venular constriction, while acute bouts of exercise can result in both arteriolar and venular dilatation [21,26,27,28]. These examples illustrate that changes in retinal vessel metrics can assess cardiovascular effects. No studies so far have investigated the effects of prenatal yoga on retinal microcirculation. The objective of this proof-of-concept study was to assess possible changes in retinal blood vessel diameters, as a proxy for microvascular reactivity, in pregnant women practicing yoga.

## 2. Materials and Methods

This observational study was performed at a single yoga studio in Maribor, Slovenia, in collaboration with the Department of Perinatology, Division of Gynecology and Perinatology, University Medical Center Maribor, Slovenia, and the Otto Loewi Research Center, Section of Physiology, Medical University of Graz, Austria, from August to October 2020.

Study participants: Participants were recruited during prenatal yoga classes from women who practiced yoga regularly at least once per week between 8 to 11 weeks and were not practicing yoga before pregnancy. Participants were attending one 90 min guided yoga class per week in the studio and encouraged to practice at home daily.

A total of 35 women were invited to be involved in the study (28 were in the second and 7 in the third trimester of pregnancy). We excluded multiple pregnancies and women with any cardiovascular, psychiatric or renal disease and those taking medications that affect heart rate or blood pressure. We also excluded women with prior ocular surgery or ocular trauma. None of the women reported tobacco smoking or vaping during pregnancy or used illicit drugs. All measurements were performed in the afternoon (same time of day). Women were not eating nor drinking coffee 1 hour prior and during the yoga class.

Yoga classes were led according to Yoga in Daily Life System [29]. They consisted of pregnancy-adapted yoga practices. The sessions lasted for 90 min and consisted of initial relaxation (10 to 15 min), followed by yoga postures (asanas) and stretching exercises (45 to 60 min), and final breathing (pranayama), concentration (dharana), and meditation (dhyana) techniques (20 to 30 min). A single certified yoga instructor with 25 years of experience in teaching pregnancy-adapted yoga led all yoga classes.

Fundus photography and retinal vessel analysis: optic disc centered images (resolution of 1536 × 1536) of both eyes were obtained 15 min before and 15 min after yoga class using a non-mydriatic, hand-held, portable 30° field-of-view digital retinal camera (Optomed Aurora, Optomed Oy, Oulu, Finland). A trained grader masked to the participant’s characteristics using semi-automated MONA REVA vessel analysis software (VITO, Boeretang, Belgium) [30]. The MONA REVA algorithm automatically segmented the retinal vessels. The segmentation algorithm is a multiscale line filtering algorithm inspired by Nguyen and coworkers [31]. Post-processing steps such as double thresholding, blob extraction, removal of small connected regions, and filling holes were performed. The diameters of the retinal arterioles and venules that passed entirely through the circumferential zone 0.5 to 1 disc diameter from the optic disc margin were calculated automatically. The trained grader verified and corrected vessel diameters and vessel labels (arteriole or venule) with the MONA REVA vessel editing toolbox. The diameters of the 6 largest arterioles and 6 largest venules were used in the revised Parr–Hubbard formula for calculating the Central Retinal Artery Equivalent (CRAE) and Central Retinal Venular Equivalent (CRVE) [32]. The results of both eyes were averaged. An example of the fundus image analysis is in Figure 1.

Statistical analysis: normality of the data was evaluated using the Shapiro-Wilk test. Continuous variables were expressed as mean ± standard deviation if normally distributed. Categorical data were summarized as frequencies and percentages. For comparison of CRAE and CRVE pre- vs. post-yoga practice, paired Student t-test was used.

Statistical significance was considered at a two-tailed *p*-value ≤ 0.05. Data were analyzed using IBM SPSS Version 28.0 (Armonk, NY, USA: IBM Corp.).

## 3. Results

All 35 women were assessed for eligibility. Six did not meet the study inclusion criteria, so 29 women were included in the study. All women were of Caucasian ethnicity and none reported tobacco smoking or vaping during pregnancy. They were all pregnant with singletons without known fetal anomaly. They did not have any chronic medical conditions, such as cardiovascular disease (including hypertension and arrhythmias), psychiatric disorders, epilepsy, kidney disease, liver disease, autoimmune disorders, thyroid disease, diabetes mellitus, alcohol and/or illicit drug abuse. The mean age of the women was 31.0 years (standard deviation (SD) ± 5.0 years, range 23–41 years). Women’s mean height was 168 cm (SD ± 6 cm, range 154–176 cm), weight 68.6 kg (SD ± 9.4, range 53–89 kg), and body-mass index 24.3 kg/m^2^ (SD ± 3.3 kg/m^2^, range 20.2–31.2 kg/m^2^). Mean gestational age was 21.6 weeks (SD ± 8.3 weeks, range 13–36 weeks).

Yoga practice did not have a statistically significant effect (*p* > 0.05) on retinal arteriolar or venular diameter (expressed as Central Retinal Arteriolar Equivalent (CRAE) and Central Retinal Venular Equivalent (CRVE), respectively), though both CRAE and CRVE values increased after yoga (Table 1). There is no correlation between the CRAE/CRVE and age, gestational age or BMI.

## 4. Discussion

We report the first feasibility study to evaluate possible microvascular effects of a yoga exercise using retinal vessels analysis. The study yielded high-quality retinal images for all participants, demonstrating the feasibility of the study setup. There were no statistically significant changes in retinal arteriolar or venular diameters in healthy pregnant women following 90 min of yoga practice. Additional studies are needed to assess the potential effects of regular yoga practice throughout pregnancy on the microvascular system.

Our finding may be due to the specific structure of the yoga sessions. Yogic physical exercises (asanas) were always followed by a relatively long period of relaxation, breathing exercises, and meditation [29]. Therefore, short-term retinal vessel vasodilatation following acute physical exercise reported in previous studies might already have elapsed by the time the measurements were performed in the present study [21]. Another possibility is that physical activity during yoga sessions might not be strenuous enough. Light physical effort makes yoga one of the most suitable forms of exercise for pregnant women but could not produce a significant acute nitric oxide-mediated retinal vasodilatation [1]. Regular physical activity leads to multiple health-promoting effects [33,34]. Knowledge of the activity patterns of individuals is important to understand vascular responses to a specific intervention aimed at inducing beneficial effects. We suggest collecting activity patterns using wearable sensors and questionnaire data. Gestational age at measurements may have also influenced the results. During pregnancy, the cardiovascular system undergoes a set of physiological changes due to increased metabolic demand of the uteroplacental fetal unit. There is a decrease in systemic vascular resistance and an increase in blood volume [35,36]. Correspondingly, there is initial decrease in systemic blood pressure, which reaches its lowest value at mid-pregnancy and begins to increase again in the last trimester to pre-pregnancy values [35,36]. Future studies will need to focus on gestational-age specific effects of physical activity, such as yoga, on the microvasculature. Nevertheless, we speculate that long-term, regular yoga practice throughout pregnancy may still result in improved microvascular reactivity, which could be one of the mechanisms by which prenatal yoga reduces risks of fetal growth restriction and hypertensive disorders of pregnancy [4,5,6,7,8,9,10,11]. Our study only examined acute changes in retinal vessel diameters immediately following yoga sessions. We do not draw any conclusions regarding the long-term effects of regular yoga practice in pregnancy on the retinal vasculature.

Our sample size was relatively small. A control group was not included; instead, we compared the effect of yoga exercise with paired measurements for each participant. Our aim was not to compare responses with controls. The literature already reports dilation of the retinal blood vessels of healthy adults in response to exercise [26,27,28]. We only investigated retinal blood vessel diameter changes before and after yoga practice and did not analyze changes in other physical activity and exercise types during pregnancy. We included participants at a single yoga studio, and a single yoga instructor led the yoga classes. The study participants attended the yoga classes voluntarily. Participation rates, i.e., how often women actually practiced yoga, could have influenced the results. They all attended one 90-min guided yoga session per week. However, we have no data on how often they practiced yoga by themselves at home. This should be viewed as the limitation of our study.

Based on the current feasibility, we plan more extensive studies, collecting physical activity data and personal characteristics such as pre-pregnancy BMI and post-partum data. Our results can be considered a baseline for retinal vessel diameters in healthy women with uncomplicated pregnancies. Our reference measurements can become valuable in future research to determine if retinal vessel analysis can assess microvascular changes due to exercise programs.

## Figures and Tables

**Figure 1 healthcare-10-01356-f001:**
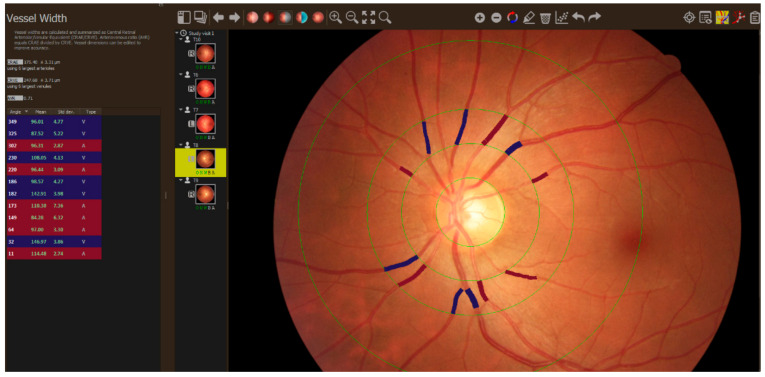
Analysis of the fundus image taken during the study.

**Table 1 healthcare-10-01356-t001:** Effect of yoga practice on retinal arteriolar and venular diameters in pregnant women (*n* = 29). Data are shown as mean ± SD; CRAE Central Retinal Arteriolar Equivalent; CRVE Central Retinal Venular Equivalent.

	Pre-Yoga	Post-Yoga	Change in Diameter	*p*-Value
CRAE (μm)	150.5 ± 11.4	151.2 ± 10.2	0.7 ± 6.0	0.53
CRVE (μm)	221.8 ± 16.1	223.2 ± 15.7	1.4 ± 11.2	0.51

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
