# Peer review of "Analysis of Retinal Blood Vessel Diameters in Pregnant Women Practicing Yoga: A Feasibility Study"

_healthcare, 2022, doi:10.3390/healthcare10071356_

Round 1

Reviewer 1 Report

The manuscript is presented in a clean and fluid style. The authors conducted the research "Analysis of retinal blood vessel diameters in pregnant women practicing yoga". This manuscript is interesting and well prepared. But, I have several questions as follows:

1. How many weeks did the participants practice Yoga? It is not clear in your study.

2. The results of the manuscript did not show significant differences in CRAE and CRVE. In fact, the rate of participation in Yoga can also affect the changes in outcomes. Therefore, the participation rates of the participants also need to be reported and explained.

Author Response

Thank you for valuable comments!

We have tried to clarify this point (1) and (2) by rewording the methods section. The paragraph now reads: “Participants were recruited during prenatal yoga classes from women who practiced yoga regularly at least once per week between 8 to 11 weeks and were not practicing yoga before pregnancy. Participants were attending one 90 minute guided yoga class per week in the studio and encouraged to practice at home daily

We agree that participation rates, i.e. how often women practiced yoga, could have influenced the results. They all attended one 90-minute guided yoga session per week. However, we have no data on how often they practiced yoga by themselves at home. This should be viewed as another limitation of our study. This has now been added in the discussion section.

Reviewer 2 Report

This manuscript is written well, so I recommend to publish this.

Author Response

Thank you for your positive evaluation of our paper!

Reviewer 3 Report

Major concerns:

1. The benefits from physical activity (both during pregnancy and overall) comes from regular exercise. The lack of this prerequisite may have greatly influenced the results

2. Authors should include more data on the subjects: pre-pregancy BMI, pregnancy complications in the study subjects, birth weight

3. The ample range of gestational age, given the small sample size, may have affected the results: vascular adaptation varies between 13 and 36 weeks of pregnancy. Authors should clarify that point

4. The lack of a control group is detrimental to the effect, but more the lack of a follow-up after regular yoga sessions

Author Response

Dear reviewer, thank you for your constructive critical opinion and comments on our work, which we have carefully analyzed.

We are providing point by point responses:

1) The benefits from physical activity comes from regular exercise. The lack of this prerequisite may have greatly influenced the results.

We agree that regular exercise, or even broader physical activity, is an important determinant of health and the expected response to an intervention. To highlight this, we have added the following clarification in the discussion section: »Regular physical activity leads to multiple health-promoting effects. Knowledge of the activity patterns of individuals is important to understand vascular compliances to a specific intervention aimed at inducing beneficial effects. We suggest collecting activity patterns using wearable sensors and questionnaire data. «

2) Authors should include more data on the subjects: pre-pregnancy BMI, pregnancy complications, birth weight…

Thank you for this valuable suggestion. The participants were recruited when they enrolled at the yoga sessions when they were pregnant. The participants did not recall specific pre-pregnancy data and we therefore decided not to include this data in our report.

We have improved the data though. In the Materials and Methods we changed the second and third paragraph that now states as follows:

Study participants: Participants were recruited during prenatal yoga classes from women who practiced yoga regularly at least once per week between 8 to 11 weeks and were not practicing yoga before pregnancy. Participants were attending one 90 minute guided yoga class per week in the studio and encouraged to practice at home daily.

Thirty-five women were invited in the study (28 were in the second and 7 in the third trimester of pregnancy). We excluded multiple pregnancies and women with any cardiovascular, psychiatric or renal disease and those taking medications that affect heart rate or blood pressure. We also excluded women with prior ocular surgery or oc-ular trauma. None of the women reported tobacco smoking or vaping during pregnancy or used illicit drugs. All measurements were performed in the afternoon (same time of day). Women were not eating nor drinking coffee one hour prior and during the yoga class.”

In the Results section we added three sentences to the first paragraph that now states as follows:

“Thirty-five women were assessed for eligibility. Six did not meet the study inclu-sion criteria. Twenty-nine women were included in the study. All women were of Caucasian ethnicity and none reported tobacco smoking or vaping during pregnancy. They were all pregnant with singletons without known fetal anomaly. They did not have any chronic medical conditions, such as cardiovascular disease (including hyper-tension and arrhythmias), psychiatric disorders, epilepsy, kidney disease, liver disease, autoimmune disorders, thyroid disease, diabetes mellitus, alcohol and/or illicit drug abuse. The mean age of the women was 31.0 years (standard deviation (SD) ± 5.0 years, range 23-41 years). Women’s mean height was 168 cm (SD ± 6 cm, range 154-176 cm), weight 68.6 kg (SD ± 9.4, range 53-89 kg), and body-mass-index 24.3 kg/m2 (SD ± 3.3 kg/m2, range 20.2-31.2 kg/m2). Mean gestational age was 21.6 weeks (SD ± 8.3 weeks, range 13-36 weeks).”

We are fully aware that pre-pregnancy and post-partum data is valuable and they will be included in future studies. We have highlight this by adding the following sentence in the discussion section: “Based on the current feasibility, we plan more extensive studies, collecting physical activity data and personal characteristics such as pre-pregnancy BMI and post-partum data.”

3) The ample range of gestational age, given the small sample size, may have affected the results: vascular adaptation varies between 13 and 36 weeks of pregnancy. Authors should clarify this point.

Yes, we agree. Thank you for this remark.

We have added a discussion on how physiological hemodynamic changes in pregnancy could have affected our results in the discussion. The section reads: ”Gestational age at measurements may have also influenced the results. During pregnancy, the cardiovascular system undergoes a set of physiological changes due to increased metabolic demand of the uteroplacental fetal unit. There is a decrease in systemic vascular resistance and an increase in blood volume. Correspondingly, there is initial decrease in systemic blood pressure, which reaches its lowest value at mid-pregnancy and begins to increase again in the last trimester to pre-pregnancy values. Future studies will need to focus on gestational-age specific effects of physical activity, such as yoga, on the microvasculature.”

4) The lack of a control group is detrimental to the effect, but more the lack of a follow-up after regular yoga sessions.

Thank you for this important remark. Please allow us to clarify this point:

Our proof-of-concept design was to demonstrate the feasibility of retinal vessel analysis in pregnant women and to investigate possible acute effects of yoga activity on the microvascular response using this retinal analysis. Although it would be beneficial, the control group was not essential to study the acute effects of yoga activity. (the 120 minutes difference between pre and post yoga class). This approach was shown valuable before in patients with heart disease and lung patients (e.g. Louwies, T.; Int Panis, L.; Alders, T.; Bonné, K.; Goswami, N.; Nawrot, T.S.; Dendale, P.; De Boever, P. Microvascular reactivity in rehabilitating cardiac patients based on measurements of  retinal blood vessel diameters. Microvasc. Res. 2019, 124, 25–29, doi:10.1016/j.mvr.2019.02.006., and Vaes, A.W.; Spruit, M.A.; Goswami, N.; Theunis, J.; Franssen, F.M.E.; De Boever, P. Analysis of retinal blood vessel diameters in patients with COPD undergoing a pulmonary rehabilitation program. Microvasc. Res. 2022, 139, 104238, doi: 10.1016/j.mvr.2021.104238.).

We highlighted in the limitation section that we did not add a control group. We mentioned the value of follow-up and collecting additional data: »Based on the current feasibility, we plan more extensive studies, collecting physical activity data and personal characteristics such as pre-pregnancy BMI and post-partum data.«

We hope that we have sufficiently addressed the reviewer's remark based on the data that we have currently available. Our future studies will definitely include this reviewer's valuable suggestions.

Round 2

Reviewer 3 Report

In this revised version, some of the major concerns have been fully addressed. Some remains, as are design flaws of the study, which may still have an impact on the results and the overall relevance.

Author Response

Dear Reviewer, 
Thank you for accepting most of our responses/modifications. 

Regarding the research design, we believe it is appropriate for a feasibility study such as this one. Please note that we have changed the title and added "a feasibility study" to emphasize this. 

I hope we have adequately explained why the control group is not essential in our research design. At the same time, we have emphasized that this is a limitation of the study and that a similar approach (without a control group) has already proven successful (citations in manuscript). 

Please let me reiterate that we are grateful for your comments and will take them into account in our future studies. 

Sincerely, Dejan Dinevski

This manuscript is a resubmission of an earlier submission. The following is a list of the peer review reports and author responses from that submission.

Round 1

Reviewer 1 Report

In the first place, the time that each woman has been practicing yoga (number of sessions, or if any woman practiced yoga before pregnancy) is not indicated.

They state that they have used the Student's t-test but do not include the results. They do not have a control group.

It would have to indicate if there is a relationship between the blood circulation of the eye and the sociodemographic data.

There are no significant differences.

Reviewer 2 Report

This is an interesting topic that unfortunately did not reach to strong conclusions due to designs limitations. Although eye small vessels have experimented changes with exercise, here the authors did not find any change with yoga. Anyway, authors must improve some issues:

Lines 22-23: although it is not relevant for results, please state the specific p-value, not the range (p>0.05)

Line 31: please state the benefits.

Line 65: selection criteria must be included here: firstly we explain about participants and latter about yoga. Please follow TIDIeR and CONSORT guidelines.

Line 71: yoga is not an official worldwide certification. How do we know the instructor has enough experience? Please include years of experience

Lines 105-106: it is not clear if 6 women withdraw along the study or were previously excluded for any reason. If this last, the sentence used is not adequate, as seems like 35 started the intervention and 6 of them withdraw.

Line 106-110: consider to use a table to facilitate readers comprehension of this information (suggestion, not compulsory)

Table 1: include p-value

References must be numbered according journal’s guidelines.